# On the Mechanism of Hyperthermia-Induced BRCA2 Protein Degradation

**DOI:** 10.3390/cancers11010097

**Published:** 2019-01-15

**Authors:** Nathalie van den Tempel, Alex N. Zelensky, Hanny Odijk, Charlie Laffeber, Christine K. Schmidt, Inger Brandsma, Jeroen Demmers, Przemek M. Krawczyk, Roland Kanaar

**Affiliations:** 1Department of Molecular Genetics, Oncode Institute, Erasmus University Medical Center, 3000 CA Rotterdam, The Netherlands; n.van.den.tempel@umcg.nl (N.v.d.T.); o.zelenskyy@erasmusmc.nl (A.N.Z.); j.odijk@erasmusmc.nl (H.O.); c.laffeber@erasmusmc.nl (C.L.); inger.brandsma@gmail.com (I.B.); 2Department of Biochemistry, The Gurdon Institute, University of Cambridge, Cambridge CB2 1QN, UK; christine.schmidt@manchester.ac.uk; 3Division of Cancer Sciences, Faculty of Biology, Medicine and Health, Manchester Cancer Research Centre, University of Manchester, Manchester M20 4GJ, UK; 4Department of Biochemistry, Erasmus University Medical Center, 3000 CA Rotterdam, The Netherlands; j.demmers@erasmusmc.nl; 5Department of Cell Biology and Histology Academic Medical Center, University of Amsterdam, 1105 AZ Amsterdam, The Netherlands; p.krawczyk@amc.uva.nl

**Keywords:** hyperthermia, homologous recombination, BRCA2, RAD51, ubiquitin, SILAC mass spectrometry, reactive oxygen species, protein degradation, HSP90

## Abstract

The DNA damage response (DDR) is a designation for a number of pathways that protects our DNA from various damaging agents. In normal cells, the DDR is extremely important for maintaining genome integrity, but in cancer cells these mechanisms counteract therapy-induced DNA damage. Inhibition of the DDR could therefore be used to increase the efficacy of anti-cancer treatments. Hyperthermia is an example of such a treatment—it inhibits a sub-pathway of the DDR, called homologous recombination (HR). It does so by inducing proteasomal degradation of BRCA2 —one of the key HR factors. Understanding the precise mechanism that mediates this degradation is important for our understanding of how hyperthermia affects therapy and how homologous recombination and BRCA2 itself function. In addition, mechanistic insight into the process of hyperthermia-induced BRCA2 degradation can yield new therapeutic strategies to enhance the effects of local hyperthermia or to inhibit HR. Here, we investigate the mechanisms driving hyperthermia-induced BRCA2 degradation. We find that BRCA2 degradation is evolutionarily conserved, that BRCA2 stability is dependent on HSP90, that ubiquitin might not be involved in directly targeting BRCA2 for protein degradation via the proteasome, and that BRCA2 degradation might be modulated by oxidative stress and radical scavengers.

## 1. Introduction

The DNA damage response (DDR) consists of various intricate pathways that maintain the integrity of the DNA in our cells, which is continuously threatened by endogenous and exogenous agents [1,2,3]. The DDR deals with DNA damage inflicted by these assaults by detecting its presence, signaling to the pathways controlling cell cycle progression, and finally repairing or bypassing the damage [2]. However, the protection provided by the DDR has justly been described as a double-edged sword—by safeguarding the DNA, it prevents accumulation of mutations in genes that might lead to cancer, but also counteracts the efficacy of many anti-cancer therapies that are based on the cytotoxicity of DNA damage [4]. It is because of the latter aspect that inhibition of the DDR is of great interest in the context of cancer treatment [5].

The importance of the DDR for treatment efficacy is illustrated by tumors harboring defects in a specific pathway of the DDR, such as homologous recombination (HR), examples of which are breast and ovarian tumors with BRCA1 and BRCA2 mutations. These tumors are not only very sensitive to specific chemotherapeutics such as cisplatin and carboplatin [6,7], but also to a new class of drugs, inhibitors of poly [ADP-ribose] polymerase 1 (PARP-1) [8,9,10]. PARP-1 inhibitors have been used to target the DDR in clinical practice [11], and are an example of precision treatment—while normal cells can compensate for a deficiency in DDR-pathways involving PARP-1-related repair by using the HR machinery, tumors that are HR-deficient cannot. Thus, the clinical efficacy of PARP-1 inhibitors in turn makes HR an attractive therapeutic target—if one could inhibit HR in a tumor, it would become more sensitive to these treatment modalities.

HR is one of the two pathways that repair DNA double-stranded breaks (DSBs), the other being non-homologous end joining (NHEJ). While, in essence, NHEJ repairs DNA by directly re-joining the two ends of a DSB together and is active throughout the cell cycle, HR faithfully restores DSBs by using an intact copy of the DNA as a template. This copy is usually the sister chromatid that arises during replication, which limits HR activity to the S and G2-phases of the cell cycle. HR is tightly regulated, and involves an orchestrated series of events mediated by many different proteins [12]. The start of the HR pathway is characterized by the generation of 3’ single-strand overhangs by DNA end resection, a process mediated by BRCA1, among other proteins. Strand invasion—a central step in the HR pathway—is catalyzed by the nucleoprotein filaments formed by RAD51 on the resected, single-stranded DNA. The assembly of RAD51 onto the DNA is aided by many proteins, including BRCA2 and RAD54 [13,14,15]. The invaded strand is now used as a template for repair of the resected strands. Resolution of the holiday junctions is the last step of HR, and can have several outcomes; however, in effect, all result in faithful repair of the broken DNA [12].

As a consequence of the complexity of HR, the pathway can be interfered with at multiple levels to create a window of therapeutic opportunities [16,17,18]. However, when thinking about such approaches, one should consider that HR is essential for cellular survival, and interfering with it could have disastrous consequences in normal cells [19]. Ideally, inhibition of HR in cancer treatment should occur in a local fashion. Hyperthermia is a method which has the potential to achieve exactly this result—by locally applying heat to a tumor, it triggers a pathway which induces degradation of BRCA2, resulting in a failure to localize RAD51 to DSBs and effectively attenuating HR [20]. BRCA2 degradation is rapidly (within ~15 minutes) induced by hyperthermia in a temperature range between 41–43 °C [21,22]. However, the molecular mechanisms of heat-mediated degradation of BRCA2 are largely unknown, with the exception of the last step which involves the proteasome [21].

Here, we investigate the mechanisms that mediate BRCA2 degradation upon heating, because this could yield: (1) a better understanding of the effects of hyperthermia; (2) an increased understanding of the modulation of the HR pathway and of the BRCA2 protein itself; (3) new therapeutic approaches to inhibit HR, and; (4) therapeutic targets that may enhance the local effects of hyperthermia.

## 2. Results

### 2.1. Heat-Mediated Degradation of BRCA2 and Modulation of HR

Hyperthermia induces BRCA2-degradation in established human cell lines of different origins, and in human tumors heated ex vivo [21,23,24]. The reduction in BRCA2 levels has been detected using the OP95 BRCA2 antibody. To confirm that the observed decline in BRCA2 protein levels is not due to changes in availability of the epitope detected by the antibody, we exposed HeLa cells expressing BRCA2, tagged with the FLAG epitope (DYKDDDDK), to 42 °C for one hour, and were able to detect a heat-induced reduction in the BRCA2 level with the FLAG-specific M2 antibody (Figure 1A).

The 3328-aa mouse BRCA2 shares 59% amino acid identity with its human orthologue [25]. To determine whether murine BRCA2 is also susceptible to degradation upon hyperthermia treatment, we exposed wild-type (IB10) and *Brca2^GFP/GFP^* [26] mouse embryonic stem cells (mES) to hyperthermia. Because the OP95 antibody has been raised against amino acids 1651-1821 of human BRCA2—a region poorly conserved in murine BRCA2 [25]—we immunoblotted with a polyclonal antibody (ab27976) recognizing the conserved N-terminus of BRCA2 and with an anti-Green Fluorescence Protein (GFP)antibody. We found that both murine BRCA2 and murine BRCA2-GFP were degraded upon hyperthermia, indicating that the heat-mediated response is evolutionarily conserved (Figure 1B).

### 2.2. Various Inhibitors Affect Heat-Mediated BRCA2 Degradation

Thermal stress generally causes protein unfolding, and it is likely that the BRCA2 protein shares this fate. Cells can deal with unfolded proteins by either refolding them with the aid of molecular chaperones, or, if a protein cannot be rescued, breaking it down. Two separate systems for protein degradation are currently recognized: the ubiquitin-proteasome pathway and lysosomal degradation (autophagy), the latter being primarily associated with processing large and aggregated proteins [27]. Heat-induced degradation of BRCA2 seems to proceed via the proteasome [21], and we could indeed confirm that the proteasome inhibitor MG132 stops this process (Figure 2A). In contrast, we found that bafilomycin A1, an inhibitor of a late step in autophagy [28], did not protect BRCA2 from degradation (Figure 2B), indicating that BRCA2 is exclusively degraded via the proteasome. 3-[3-(Cyclopentylthio)-5-[[[2-methyl-4′-(methylsulfonyl)[1,1′-biphenyl]-4-yl]oxy]methyl]-4H-1,2,4-triazol-4-yl]-pyridine, 3-[3-Cyclopentylsulfanyl-5-(4′-methanesulfonyl-2-methylbiphenyl-4-yloxymethyl)-[1,2,4]triazol-4-yl]-pyridin (NMS-873), which inhibits valosin-containing protein (VCP; also known as p97 or CDC48), a “segragase” acting in many cellular processes—including extraction of ubiquitinated proteins from chromatin and membranes [29]—also protected BRCA2 from degradation (Figure 2C). BRCA2 is thought to be a client protein of the molecular chaperone HSP90, and a prolonged inhibition of HSP90 causes reduction in BRCA2 levels [30]. As expected, inhibition of HSP90 indeed enhanced BRCA2 degradation upon hyperthermia (Figure 2D) [21,31]. To test the importance of renewed synthesis of BRCA2 during hyperthermia treatment, we treated cells with the translation inhibitor cycloheximide. Surprisingly, we found that inhibition of translation protected BRCA2 from degradation (Figure 2E). This protection expired after two hours of treatment, which indicates that translation is important to maintain BRCA2 protein levels (Figure 2F).

To determine whether or not the fraction of BRCA2 protected by the different inhibitors was still functional, we tested for focus formation of the protein product of the Radiation 51 RAD51 gene (Figure 2G). Formation of RAD51 foci upon irradiation is strictly dependent on BRCA2 [13,32] and has been demonstrated to be abrogated by heat [21,22]. Both MG132 and VCP inhibition impair RAD51 focus formation [33,34], so we were able to perform this assay for cycloheximide only. Cycloheximide decreased the number of RAD51 foci in irradiated cells incubated at 37 °C, but treatment with heat failed to enhance this effect, indicating that the abundance of BRCA2, as well as its functionality might be protected by inhibition of translation.

### 2.3. BRCA2 Stability Is Dependent on HSP90

Targeting HSP90 by inhibitors like 17-Dimethylaminoethylamino-17-demethoxygeldanamycin (17-DMAG) increase the extent of BRCA2 degradation upon hyperthermia [21]. We have shown that a short exposure to the second-generation HSP90-inhibitor, ganetespib, achieves the same result and increases the cells’ sensitivity to hyperthermia (Figure 2D) [31]. To further explore the extent by which BRCA2-protein stability is dependent on HSP90 upon being challenged by hyperthermia, we combined cycloheximide, ganetespib, and hyperthermia in a single experiment. We found that the protective effects of cycloheximide on BRCA2 were at least partly dependent on HSP90 (Figure 3A). Moreover, we found that the autophagy inhibitor bafilomycin did not rescue BRCA2 from the combination of hyperthermia and ganetespib, while MG132 did (Figure 3B). This indicates that BRCA2 is cleared by the proteasome in the presence of an HSP90 inhibitor and hyperthermia, as is the case for hyperthermia alone (Figure 2A) [21]. All these results combined reinforce the hypothesis that HSP90 protects BRCA2 under hyperthermic conditions.

### 2.4. Searching for the Proteins that Mediate Degradation of BRCA2 upon Hyperthermia

The cascade of events that leads to proteasomal degradation of BRCA2 upon heat treatment might involve (unknown) BRCA2 interaction partners, which could be interesting from both a biological and clinical perspective. Usually, a protein is post-translationally marked with ubiquitin before being degraded by the proteasome. Ubiquitination is mediated by a number of enzymes [35]. First, an E1-activating enzyme activates and transfers a ubiquitin molecule to the E2-conjugating enzyme. The E2-enzyme is recruited to an E3-ubiquitin ligase, which catalyzes the transfer of ubiquitin to a target lysine of the protein. Being the last in the ubiquitin-cascade, the E3-ligases provide the specificity to the target protein, and are therefore potentially the most interesting interaction partners. However, over 600 different E3-ligases have been identified thus far [36] and pinpointing the E3-ligases responsible for heat-mediated BRCA2 degradation is challenging. The number of known E2-conjugating enzymes is more manageable. Moreover, E2 enzymes can be used to identify their downstream partner, E3 ligase(s) [37]. Therefore, we set out to individually knock down all E2-conjugating enzymes by siRNA in U2OS cells (Figure 4A). However, downregulation of no single E2-conjugating enzyme protected BRCA2 from heat-induced degradation (Figure 4A). To test for redundancy between E2-enzyme families, we combined siRNAs to simultaneously downregulate multiple E2-enzymes, but found no effect on BRCA2 levels (Figure 4B). The small ubiquitin-like modifier (SUMO) is also able to target proteins to the proteasome, and SUMOylation is increased upon heat-shock [38]. We therefore downregulated the E2-conjugating enzyme for SUMO and UBC9 (also known as UBE2I) [39], but found that this also had no effect on BRCA2 protein levels after hyperthermia (Figure 4C).

Because no candidate E2-conjugating enzymes emerged from the systematic screen, we decided to try a candidate approach for several E3-ligases. The first E3-ligase we included, C terminus of HSC70-Interacting Protein (CHIP; also known as STUB1), is involved in the heat-shock response [40], cooperating with multiple E2-conjugating enzymes [41], and has been implicated in the degradation of the base excision repair protein 8-Oxoguanine glycosylase (OGG1) upon hyperthermia [42]. However, knockdown of CHIP did not affect BRCA2 degradation upon heat treatment (Figure 5A). Knockdown of the four SUMO E3-ligases Protein Inhibitor Of Activated STAT (PIAS) 1-4 [43] did not affect BRCA2 protein levels (Figure 5B), and nor did the knockdown of two SUMO-targeted ubiquitin ligases (STUbLs), RNF4 and RNF111 (also known as Arkadia), both implicated in DNA double-strand break (DSB) signaling [39,44] (Figure 5C). We found that the absence of BRCA1, which is closely related to the BRCA2 complex and which has E3-ligase activity [45,46,47], did not change degradation of BRCA2 upon hyperthermia (Figure 5D). Inhibition of the neddylation 8 (NEDD8)-activating Enzyme 1 (NAE1), which prevents neddylation and thereby activation of all Cullin-type E3 ligases [48,49], also did not affect BRCA2 degradation (Figure 5E). Finally, we abolished all ubiquitination by inhibiting the E1-activating enzyme using PYR-41 [50] and, surprisingly, found that addition of this inhibitor failed to protect BRCA2 upon applying hyperthermia, in contrast to the addition of a proteasome inhibitor (Figure 5F).

### 2.5. Heat-Mediated BRCA2 Degradation from the Protein’s Perspective

Since neither the E2-wide screen nor the E3-candidate approach yielded proteins involved in mediating hyperthermia-induced BRCA2 degradation, we decided to approach the problem from the perspective of BRCA2 itself (Figure 6A). This protein has a number of properties which complicate various proteomic approaches in the presence of hyperthermia: BRCA2 is (1) large (384 kDa); (2) expressed at a relatively low level and only during the S and G2 phase of the cell cycle [51,52]; (3) predicted to have many intrinsically disordered domains [53], and; (4) degraded upon treatment with hyperthermia.

In order to side-step some of the properties listed above, we started by engineering three fragments of *BRCA2*: a fragment starting from the N-terminus (amino acids 1-939), a middle part (amino acids 940-2198), and a fragment ending in the C-terminus (amino acids 2199-3418) (Figure 6B). We tagged them with either GFP (Figure 6C) or FLAG (Figure 6D) and introduced them to HeLa cells using a PiggyBac transposon system [54]. We found that all GFP-tagged versions of the BRCA2-fragments displayed some degradation upon hyperthermia, although at much reduced levels compared to endogenous BRCA2 (Figure 6C). The degradation was most pronounced for BRCA2-Middle (Figure 6C), which we also found when using a FLAG-tagged version (Figure 6D). We engineered a fourth FLAG-construct, encoding the fused N-terminal and C-terminal parts of BRCA2 (ΔMiddle), and found that this fragment was also degraded upon heat, in contrast to the separate FLAG-tagged N-terminal or C-terminal fragments (Figure 6D). This observation, together with the difference in the extent of degradation of BRCA2-fragments tagged with either GFP or FLAG, suggest that expression levels of the constructs could potentially influence the results in this assay—overexpression of the fragments may saturate the molecular components which mediate degradation.

To obtain more conclusive evidence for which part of BRCA2 is degraded upon hyperthermia, we took advantage of the behavior of endogenous BRCA2. Since the protected BRCA2 is shifted to the insoluble fraction of cell lysates [22], we subjected cells expressing BRCA2 fragments to hyperthermia in the presence of MG132 and performed a simple fractionation using a mild detergent (Figure 6E). We found that the FLAG-tagged full-length protein behaved identically to the untagged wild-type BRCA2. Hyperthermia caused the BRCA2-protein to shift from a soluble fraction, and addition of MG132 dramatically increased its presence in the insoluble fraction (Figure 6E) [22]. The same localization of BRCA2 protein fragments in the supernatant and pellet was found in cells expressing BRCA2-Middle, BRCA2-C-terminus, and BRCA2ΔMiddle (Figure 6E). Interestingly, the N-terminal version of BRCA2 did not disappear from the supernatant upon treatment with hyperthermia alone, but did so when MG132 was added (Figure 6E). Together, these results suggest that the Middle and C-terminal fragments contain the major determinants for heat-mediated BRCA2 degradation, and might therefore be of interest to identify heat-dependent interaction partners.

### 2.6. Semi-Quantitative Mass Spectrometry Analysis Identifies Putative BRCA2-Interactors upon Hyperthermia

In parallel to the previous approach, and as another way to investigate hyperthermia-mediated degradation of BRCA2 from the protein’s perspective, we performed a proteomic analysis in *Brca2^GFP/GFP^* mES cells [26], where BRCA2 was immunoprecipitated using GFP nanobody beads. A three-state reciprocal Stable Isotope Labelling with Amino acids in Cell culture (SILAC) approach was used to quantitatively compare three different treatments: 0, 20, and 60 min of hyperthermia at 42 °C. Log_2_ SILAC ratios were calculated for each protein, and revealed that 20 min exposure to hyperthermia enriched the immunoprecipitate of BRCA2-GFP for Ubiquitin, USP28, and HSPB1 (also known as HSP25 and HSP27) (Table 1). HSPB1 and USP28 remained enriched in the immunoprecipitate after 60 min of hyperthermia (Table 1). Hyperthermia induces BRCA2 degradation, and therefore the decreased abundance of BRCA2, and known interactors such as RAD51 and PALB2 after 60 min at 42 °C can be regarded as a quality control within this experiment (Table 1).

### 2.7. Oxidative Stress Induces BRCA2 Degradation

The proteomic experiment identified HSPB1 as a prominent addition to the BRCA2 interactome shortly after hyperthermia (Table 1) and also confirmed the previously published observation that Kelch-like ECH-associated protein 1 (KEAP1) is part of the BRCA2 complex [47]. Interestingly, both proteins participate in the oxidative stress response [55,56]. This is of interest, because hyperthermia is thought to alter the redox state of cells by inducing an imbalance between the production of reactive oxygen species (ROS) and the presence of radical scavengers [57]. For instance, hyperthermia increases ROS by upregulating nicotinamide adenine dinucleotide phosphate oxidase (NADPH oxidase) [58] and has been implicated in downregulation of the antioxidant glutathione [59]. This prompted us to investigate whether there was a connection between oxidative stress and BRCA2 protein stability.

To test the possible relation between heat-induced oxidative stress and BRCA2 degradation, we first analyzed whether introduction of exogenous antioxidants could protect BRCA2 from being degraded by hyperthermia. We found that the anti-oxidants N-acetylcysteine (NAC) [60] and ascorbic acid at low concentrations protected BRCA2 in hyperthermic circumstances (Figure 7A). Dithiothreitol (DTT), which reduces disulphide bonds of proteins, did not prevent BRCA2 degradation upon hyperthermia, but lowered protein levels upon increased concentration under non-hyperthermic conditions (Figure 7A).

Previously, it was found that ultraviolet B irradiation (UV-B) causes downregulation of BRCA2, which was not dependent on cell cycle differences and could be blocked by cycloheximide [61]. In the context of the relation between oxidative stress and BRCA2, this is of special interest because it has been reported that UV-B induces ROS [62,63] and potentiates a disturbance in oxidative homeostasis. We found that UV-B indeed downregulated BRCA2 (Figure 7B). Next, we added several compounds that increased the amount of ROS in the cell, further disturbing the oxidative balance. While the mitochondrial complex I inhibitor rotenone [64] and the organic peroxide tert-Butyl hydroperoxide (t-BHP) did not clearly decrease BRCA2 protein levels, addition of H_2_O_2_ did (Figure 7C). Like hyperthermia-induced degradation of BRCA2, inhibition of the proteasome protected BRCA2 from H_2_O_2_-mediated degradation (Figure 7D). Failure of rotenone to induce BRCA2 degradation can be explained by mitochondrial localization of the ROS it produces. Organic (t-BHP) and inorganic (H_2_O_2_) peroxides are metabolized differently, and can have profoundly different effects on DNA integrity, poly (ADP-ribose) polymerase (PARP) activation, and apoptosis induction [65,66,67]. In the context of the relation between oxidative stress and BRCA2, this is of special interest because it has been reported that UV-B induces ROS [61,62] and potentiates a disturbance in oxidative homeostasis. We found that UV-B indeed downregulated BRCA2 (Figure 7B). Next, we added several compounds that increased the amount of ROS in the cell, further disturbing the oxidative balance. While the mitochondrial complex I inhibitor rotenone [63] and the organic peroxide *tert*-Butyl hydroperoxide (t-BHP) did not clearly decrease BRCA2 protein levels, addition of H_2_O_2_ did (Figure 7C). Like hyperthermia-induced degradation of BRCA2, inhibition (Figure 7D) of the proteasome protected BRCA2 from H_2_O_2_-mediated degradation (Figure 7D). Failure of rotenone to induce BRCA2 degradation can be explained by mitochondrial localization of the ROS it produces. Organic (t-BHP) and inorganic (H_2_O_2_) peroxides are metabolized differently, and can have profoundly different effects on DNA integrity, PARP activation, and apoptosis induction [64,66]. Which of these differences is responsible for the distinct effects on BRCA2 requires further investigation.

## 3. Discussion

### 3.1. Heat-Mediated Inhibition of HR from an Evolutionary Perspective

In the context of cancer treatment, hyperthermia is an effective method to increase anti-cancer efficacy of DNA-damaging treatments, such as radiotherapy and some forms of chemotherapy [67]. However, in the 19th century, hyperthermia was applied as an anti-cancer treatment in a substantially different way—patients were treated with bacterial toxins that induced fever [68,69]. This treatment can be regarded as hyperthermia treatment, as well as immunotherapy—and underlining that fever, an increase in the core temperature of an organism, is a physiological situation during which hyperthermia occurs. Interestingly, fever is conserved in both warm-blooded and cold-blooded vertebrates, and has several very important effects on the immune system [70]. Considering fever and the fever-regulated response from this evolutionary and physiological perspective sheds a different light on the relevance of BRCA2 degradation upon hyperthermia. It could be speculated that it is not the downregulation of BRCA2 in the fever temperature range that is important per se, but the functional effect of it—attenuation of HR.

From an evolutionary point of view, downregulating HR during fever could have an advantage for the organism. Fever is part of the immune response that usually occurs when a foreign particle has infiltrated the body, and we could therefore hypothesize that a concurrent downregulation of HR might protect the organism against these foreign particles. Indeed, it is known that some viruses hijack DNA damage repair pathways to replicate their own genomes, and there are examples of viruses, such as Epstein-Barr and human papillomaviruses, which employ HR proteins to do so [71,72,73]. However, for now, this hypothesis remains a mere speculation. It would be interesting to examine whether BRCA2 orthologues other than humans and mice, such as primates [74] or chicken [75], are also degraded upon heat treatment. The latter species could yield valuable information about this possibly conserved response, because birds have a higher basal body temperature than mammals, but can still get fever [76].

### 3.2. BRCA2 Protein Stability, HSP90, and Cycloheximide

BRCA2 is expressed in the S and G2 phases of the cell cycle, and BRCA2-transcription peaks at the start of the S phase [51,52]. Although the half-life of BRCA2 is not precisely known, it is clear that it is severely shortened by heat. The exact difference in this half-life cannot be determined by the straightforward application of cycloheximide, as this translation inhibitor protects BRCA2 from degradation. There are at least two explanations for this effect. Firstly, cycloheximide prevents translation of a certain factor which is rapidly synthesized upon hyperthermia, which could be necessary for mediating BRCA2 degradation. Secondly, cycloheximide prevents synthesis of many nascent polypeptides that rely on HSPs to be properly folded, thereby allocating more HSPs to chaperone unstable proteins, including BRCA2 [77]. Given that the protection provided by cycloheximide is partly dependent on HSP90 (Figure 3A), the latter hypothesis seems more likely.

### 3.3. HSP90 Is an Attractive Target to Modulate Heat-Induced BRCA2 Degradation

Hyperthermia-mediated degradation of BRCA2, localization defects of RAD51, and sensitivity to irradiation are all enhanced by the addition of 17-DMAG or ganetespib [21,31]. There is evidence that HSP90 influences the steady-state level of BRCA2 by affecting both BRCA2 synthesis as well as degradation in hyperthermic conditions. The previous demonstration that BRCA2 levels decline when cells are treated with an HSP90-inhibitor for a prolonged period of time, indicates that HSP90 is important for BRCA2 homeostasis [30]. By combining cycloheximide with ganetespib, we provide evidence for the importance of HSP90 in preventing BRCA2 degradation. The BRCA2 protein signal observed after treatment with cycloheximide represent molecules, which existed prior to treatment with hyperthermia and ganetespib, dramatically reduces these protected BRCA2 molecules in hyperthermic conditions (Figure 3A). Thus, HSP90 is not only important for the proper folding of newly synthesized BRCA2, but also protects unfolding molecules from being degraded. Therefore, HSP90 is currently among one of the most attractive targets to modulate heat-induced BRCA2 degradation and to increase hyperthermia treatment efficacy.

### 3.4. Ubiquitination May Not Be Required for Heat-Mediated BRCA2-Degradation by the Proteasome

BRCA2 degradation upon heat is mediated by the proteasome, and the most common pathway for targeting proteins to the proteasome involves the ubiquitin system [78]. Indeed, we demonstrated that inhibition of the ubiquitin-selective valosin-containing protein (VCP) segregase prevents degradation, and we found ubiquitin enrichment in immunoprecipitates of the BRCA2-complex shortly after hyperthermia induction (Table 1), both of which are indications that BRCA2 degradation might be regulated via ubiquitination. However, our efforts to determine which ubiquitin enzymes might be involved in this process were not successful (Figure 4 and Figure 5). One E3-ligase we have not specifically tested as a candidate, but which might be important in BRCA2 regulation, is KEAP1. This protein is detected in the proteomic screen as an interactor of BRCA2, and has previously been associated with the complex [47]. However, we did not observe any difference in efficiency of BRCA2 degradation when we added the neddylation inhibitor MLN4924, which inhibits all cullin-based E3-ligases, including KEAP1 [46,79] (Figure 5E).

The negative results of the E2- and E3-screens can be interpreted in a number of ways. Our screens are predominantly based on short-interfering RNA, and insufficient knock-down could therefore be a problem, considering how low levels of E2 or E3 proteins might be enough to target the lowly expressed BRCA2 to degradation. Moreover, E2 and E3 enzymes may be redundant. For example, our finding that both the middle- and C-terminal parts of the BRCA2 protein are heat-labile opens up the possibility that different E3 enzymes initiate degradation by targeting distinct parts of BCRA2 (Figure 6). However, the key experiment that may explain the inability to find a responsible E2 or E3 ubiquitin enzyme is that inhibition of the E1 enzyme [50], which effectively inhibits the entire ubiquitin system, still results in degradation of BRCA2 (Figure 5F). Thus, ubiquitination may actually not be directly involved in hyperthermia-mediated BRCA2 degradation.

Although most proteins are ubiquitinated before being targeted to the common 26S proteasome form, there is an exception to this rule—proteins may interact with the 26S proteasome via a ubiquitin-like domain, as is the case for RAD23B [80]. Similarly, it might be that conformational changes of BRCA2 induced by hyperthermia result in exposure of such a ubiquitin-like domain. There are other examples of ubiquitin-independent degradation of proteins, but they require a different proteasome composition. The classic 26S proteasome consists of two 19S regulatory particles which provide ubiquitin-specificity, while final degradation takes place in the 20S core particle [81]. The assembly of the proteolytic core itself may change to form the immunoproteasome, which has different proteolytic activity than the classic 20S core, enabling production of peptides for antigen presentation [82]. Formation of the immunoproteasome is stimulated by the γ-interferon as part of the adaptive immune response. The proteolytic 20S core can act on its own or be regulated by particles other than the 19S subunit—i.e., the PA200 and the 11S/PA28αβ subunits [82,83]. The PA200 regulatory subunit is specialized in degrading acetylated histones, and is therefore an example of a ubiquitin-independent form of the proteasome [84]. The PA28αβ regulators are involved in the degradation of oxidatively damaged proteins, as is the immunoproteasome [85]. Examples of proteins that may be degraded by the 20S proteasome on its own are proteins with intrinsically disordered domains which are degraded “by default” [86], or proteins which are oxidized [87].

Several properties of BRCA2 make it a likely candidate for degradation by a ubiquitin-independent proteasome. For instance, the ubiquitin-independent degradation “by default” pathway, mediated by the 20S proteasome, is hypothesized to act on proteins that expose intrinsically disordered domains after losing interactions with other components of the protein complex in which they normally reside [86]. BRCA2 is indeed predicted to have intrinsically disordered domains, and may drastically change conformation upon temperature changes [53]. Furthermore, well-known interactors of BRCA2, PALB2 [88,89], and RAD51 [26,90] are not degraded upon heat (Figure 1D) [21], suggesting that BRCA2 is released from complexes with these proteins before being degraded. Oxidative stress may also be important for heat-mediated BRCA2 degradation via the 20S proteasome, immunoproteasome, or PA28αβ proteasomes, as illustrated by the effects of oxidative stress on BRCA2 stability (Table 1).

Experimental possibilities to distinguish between the 20S proteasome and the ubiquitin-dependent 26S proteasome are limited. Proteasome inhibitors, including MG132, inhibit both the 20S and 26S proteasome pathways. However, one way to distinguish between 20S and 26S proteasome activity is by depleting Adenosine triphosphate (ATP), because the 26S is dependent on ATP, while the 20S is not [91]. This may, however, have severe consequences for cell viability in general, and may therefore not be an optimal approach. Other experiments that could elucidate the type of proteasome required for heat-mediated BRCA2-degradation involve selectively knocking down or inhibiting the immunoproteasome, the 19S regulatory particle, or the PA28αβ subunit, which can be achieved by siRNA [85,91].

### 3.5. BRCA2 and the Oxidative Stress Response

In the proteomic screen aimed at the identification of BRCA2-interactors upon hyperthermia, we identified HSPB1 as an interactor which dramatically increases in abundance shortly after hyperthermia induction. We also identified KEAP1 as a constitutive part of the complex (Table 1). Interestingly, these proteins are currently considered to function in the oxidative stress response. Since cellular response to hyperthermia and oxidative stress have many similarities [57,92,93], and since important players of the HR machinery have been associated with the oxidative stress response [47,94], we performed a screen with a limited number of ROS-inducing agents, as well as antioxidants, and analyzed their effects on BRCA2. Our data indicate that the level of BRCA2 is in fact decreased in response to oxidative stress, while radical scavengers can protect BRCA2, to a certain extent, from hyperthermia-mediated degradation (Figure 7). It is tempting to speculate that hyperthermia-induced BRCA2 degradation is at least partly dependent on oxidative stress. Notably, BRCA1 and PALB2, both well-established BRCA2-interactors, are involved in the oxidative stress response [47,94]. Oxidative stress may cause dissociation of these proteins from the complex, resulting in destabilization of BRCA2.

## 4. Materials and Methods

### 4.1. Cell Culture

HeLa, BRO, and U2OS cells were maintained as previously described [21], as were BRCA2^GFP/GFP^ and the parental wild-type (IB10) mouse embryonic stem cells (mES) [26]. The human ovarian carcinoma cells UWB-1.289 with and without wtBRCA1 were kindly provided by Dr. Helleman of the Department of Medical Oncology of the Erasmus University Medical Center, and cultured as described [95].

### 4.2. Generation of Constructs and Cell Lines

A full list of constructs and oligonucleotides used in this study can be found in Table 2 and Table 3. A step-wise description of engineering the GFP-BRCA2 encoding plasmid has been provided previously [26]. The BRCA2 fragments used in this study were engineered in a similar fashion—stepwise Gibson assembly [96] was used to engineer the pGb-LPL vector by replacing the gene trap cassette of a 5’-PTK-3’ PiggyBac vector [54] with a puromycin acetyltransferase expression cassette with a phosphoglycerate kinase (PGK) promotor, and a bovine growth hormone polyadenylation signal from pCAGGS-Dre-IRES-puro [97]. GFP-tagged constructs were engineered by inserting the following elements into the pGb-LPL vector: a BRCA2-expression cassette consisting of a CAG promotor, consisting of (**C**) the cytomegalovirus (CMV) early enhancer element, (**A**) the promoter, the first exon and the first intron of chicken beta-actin gene, (**G**) the splice acceptor of the rabbit beta-globin gene, from pCAGGS-Dre-IRES-puro [97], an Enhanced Green Fluorescence Protein (EGFP)-coding sequence (Takara Bio), and the indicated BRCA2 fragment (using primer set 1, 2 and 3), amplified from a human BRCA2-coding sequence, phCMV1-MBPx2-hBRCA2 [98]. FLAG-BRCA2 sequences were engineered by inserting the SV40-polyadenylation signal and cytomegalovirus (CMV)-sequence from phCMV1-MBPx2-hBRCA2, as well as a 3xFLAG sequence from pR6K-2T1-2PreS-mVenus-Biotin-T2A-gb3-neo, and the indicated BRCA2 fragment (using primer sets 4, 5, and 6) in the pGb-LPL vector. The FLAG-BRCA2 construct was generated by excising the eGFP-fragment from the GFP-BRCA2-plasmid using uniquely cutting restriction enzymes and restoring the excised part with a PCR-generated patch (using primer set 7) containing the 3xFLAG-sequence and part of the N-terminus of BRCA2 from FLAG-BRCA2-Nterm. FLAG-BRCA2ΔM was generated by excising the sequence coding for the middle part using uniquely cutting restriction enzymes. The sequence adjacent to the middle part that was included in excision, was restored by a PCR-generated patch (using primer set 8) from the original construct. All PCR-fragments amplified during plasmid engineering, as well as the cloning junctions, were verified by sequencing. As described previously for the GFP-BRCA2 construct [26], stable cell-lines were created by co-transfecting 1 µg of construct with 1 µg PiggyBac transposase expression construct (mPB) per 6-well plate, using the X-tremeGENE HP DNA transfection reagent (Roche). 1 d after transfection, the selection process began by exposure to 1.5 µg/mL puromycin (Invivogen) for 10 days. The resulting puromycin-resistant cells were assayed as a mixed population.

### 4.3. Hyperthermia Treatment

The time designated for effective hyperthermia treatment started 15 min (referred to as pre-heating time) after moving the cells to an incubator set at 42 °C. Unless stated otherwise, cells were treated with 60 min of hyperthermia.

### 4.4. Chemical Agents and UV Irradiation

Cells were UV-irradiated with Ultraviolet (UV)-B (2 TL-12 (40W) tubes, Philips) at the indicated doses. A full list of used chemical agents and suppliers can be found in Table 4. Unless stated otherwise, inhibitors were added to the medium 30–60 min prior to hyperthermia and left on the cells for the duration of the treatment. Cells were treated with antioxidants and oxidative-stress-inducing agents in Hank’s Balanced Salt Solution (Gibco ® HBSS, Thermo Fisher). Antioxidants were added directly prior to hyperthermia, while treatment with the oxidative-stress-inducing agents lasted 75 min.

### 4.5. siRNA Transfection

A full list of used siRNAs with suppliers and references can be found in Table 5. Cells were transfected with a final concentration of 60 nM siRNAs using Lipofectamine RNAiMAX (Life Technologies). The systematic E2 screen was performed as previously described [37]. Experiments were performed 48–72 h after transfection.

### 4.6. Immunoprecipitation

For each immunoprecipitation, cells were prepared in a 15 cm dish at 70–80% confluency. After treatment, cells were washed twice in ice-cold PBS and lysed for 30 min on ice in NETT buffer (100 mM NaCl, 50 mM Tris pH 7.5, 5 mM ethylenediaminetetraacetic acid (EDTA) pH 8.0, 0.5% Triton-X100, 1x protease inhibitors (Complete, Roche®) and 1 mM pefabloc). Anti-Green Fluorescence Protein (GFP) beads (Chromotek) were prepared according to the manufacturer’s protocol. After scraping the cells, the suspensions were centrifuged (12000 rcf, 15 min), and the resulting supernatant (input) was added to the beads and incubated for 4 h, rotating at 4 °C. The immunoprecipitation was completed according to the manufacturer’s protocol.

### 4.7. Cell Fractionation, Lysis, and Immunoblotting

The cell fractionation was performed as described for immunoprecipitations. The pellet and supernatant resulting from the centrifugation step were analyzed. Cell lysis and immunoblotting were performed as described previously [22], with the exception of gel type, which included acrylamide, 3–8% Tris-Acetate gels (Novex, Thermofisher Scientific) or 4–20% TGX^TM^-gels (Biorad), and blotting membrane, which included both PVDF and nitrocellulose.

### 4.8. Immunofluorescent Staining and Analysis of RAD51-Foci

Cells were treated with hyperthermia and subsequently irradiated with a caesium-137 source with a dose rate of 0.64 Gy/min. EdU (Invitrogen) was added to the cells 45 min prior to fixation to identify S-phase cells. Cells were fixed 1.5 h after irradiation. Cell fixation, immunofluorescent staining, and image acquisition and analysis were performed as described in [22].

### 4.9. Antibodies

A full list of antibodies and used dilutions can be found in Table 6.

### 4.10. SILAC-Based Mass Spectrometry

Cells were cultured in DMEM medium without lysine or arginine, and supplemented with dialyzed serum for two weeks prior to the experiment. Isotopes of L-lysine and L-Arginine (Cambridge Isotope Laboratories) were added to the medium to label the following three SILAC-states: Light (K0R0), Medium (K4R6), and Heavy (K8R10). Immunoprecipitations were performed for each state separately, as described above. As published previously [101], bound proteins of all three states were mixed after immunoprecipitation and digested “on bead”. Data were analyzed using the Andromeda Search Engine within the MaxQuant software package, version 1.5.3.8 [102,103].

## 5. Conclusions

Hyperthermia attenuates HR by downregulating HR proteins, including BRCA2, thereby rendering innately HR-proficient cells sensitive to PARP inhibitors. Understanding the mechanism by which hyperthermia targets BRCA2 for degradation is interesting from both a biological and clinical perspective, because it could yield a general understanding of hyperthermia biology, regulation of BRCA2 and HR, and eventually unravel other therapeutic targets that may inhibit HR or increase hyperthermia efficacy [67]. Here, we explored this mechanism and presented molecular details of hyperthermia-mediated degradation of BRCA2. With regard to BRCA2, we showed that its heat-induced degradation is evolutionarily conserved, that its stability is guarded by HSP90, that ubiquitin might not be involved in directly targeting it for protein degradation via the proteasome, and that its degradation might be modulated by oxidative stress and radical scavengers.

## Figures and Tables

**Figure 1 cancers-11-00097-f001:**
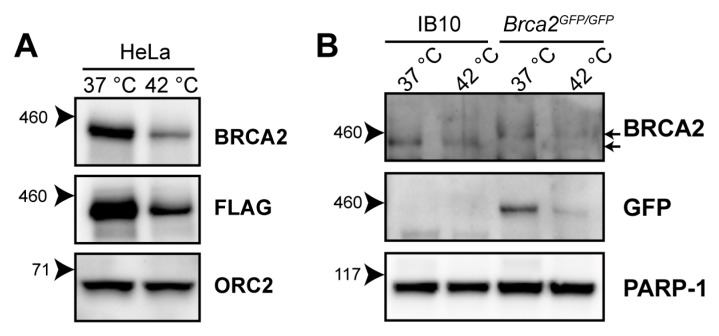
Heat-mediated degradation of BRCA2 and modulation of homologous recombination (HR). (**A**) Immunoblot of HeLa cells stably expressing expressing BRCA2, tagged with the FLAG epitope (DYKDDDDK), treated with or without 60 min of hyperthermia. (**B**) Immunoblot of wild-type IB10 and BRCA2^GFP/GFP^ mouse embryonic stem (mES) cells. The upper and lower arrow next to the upper BRCA2 panel indicate the positions of BRCA2-GFP and untagged BRCA2, respectively.

**Figure 2 cancers-11-00097-f002:**
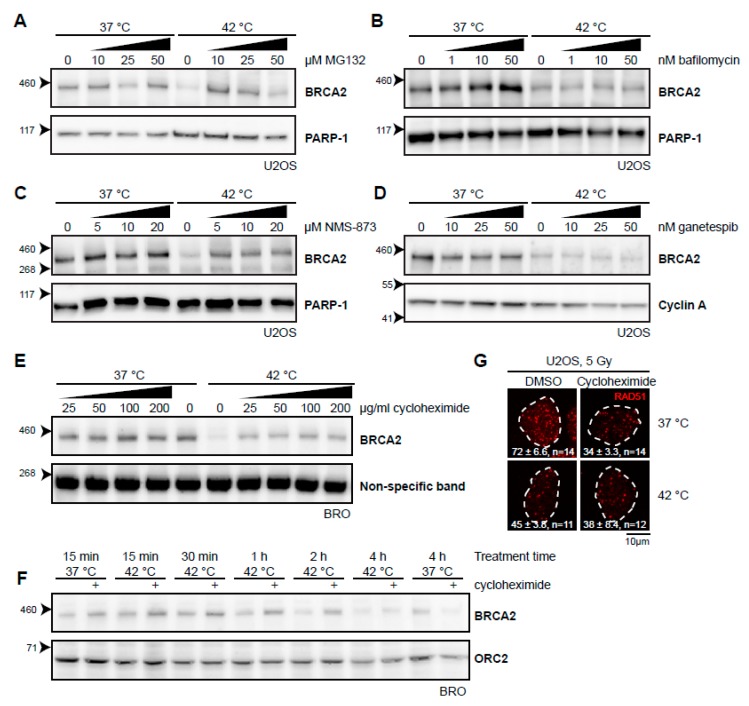
Various inhibitors alter the heat-mediated BRCA2 degradation. (**A**–**E**) Immunoblots of cells treated with or without 60 min of hyperthermia in the presence of indicated doses of different inhibitors. All inhibitors were added 30–60 min prior to hyperthermia treatment. (**A**) Human Bone Osteosarcoma U2OS cells treated with the proteasome inhibitor MG132. (**B**) HeLa cells treated with the autophagy inhibitor bafilomycin A1. (**C**) U2OS cells treated with an inhibitor of the valosin-containing protein (VCP) segregase. (**D**) U2OS cells treated with the HSP90 inhibitor, ganetespib. (**E**) Lymphoma-derived BRO cells treated with the translation inhibitor, cycloheximide. (**F**) BRO cells treated with cycloheximide (50 µg/mL) and hyperthermia for the indicated periods of time. (**G**) U2OS cells were treated with or without hyperthermia in the presence of dimethylsulfoxide (DMSO) and cycloheximide, irradiated with 5 Gy, and fixed 90 min after irradiation. Cells were stained for 5-ethynyl-2’-deoxyuridine (EdU) and the protein product of the Radiation 51 RAD51 gene. The panel shows the RAD51-staining for representative cells for each condition. The dotted line indicates the perimeter of an EdU-positive nucleus. Numbers in the panel indicate the mean number of foci ± standard error of the mean and the number of cells analyzed.

**Figure 3 cancers-11-00097-f003:**
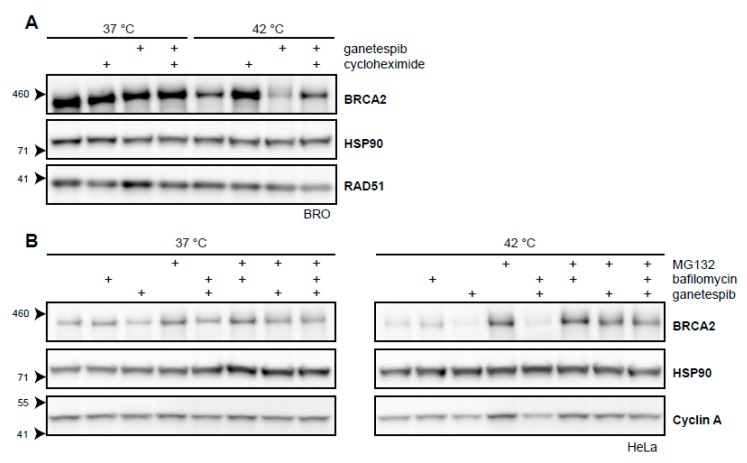
BRCA2 is a client protein of HSP90. (**A**) Immunoblot of BRO cells treated with or without 60 min of hyperthermia in the presence of cycloheximide (50 µg/mL), ganetespib (50 nM), or both. (**B**) Immunoblots of HeLa cells treated with or without hyperthermia in the presence of various combinations of proteasomal inhibitor MG132 (50 µM), autophagy inhibitor bafilomycin (10 nM), and HSP90-inhibitor ganetespib (50 nM).

**Figure 4 cancers-11-00097-f004:**
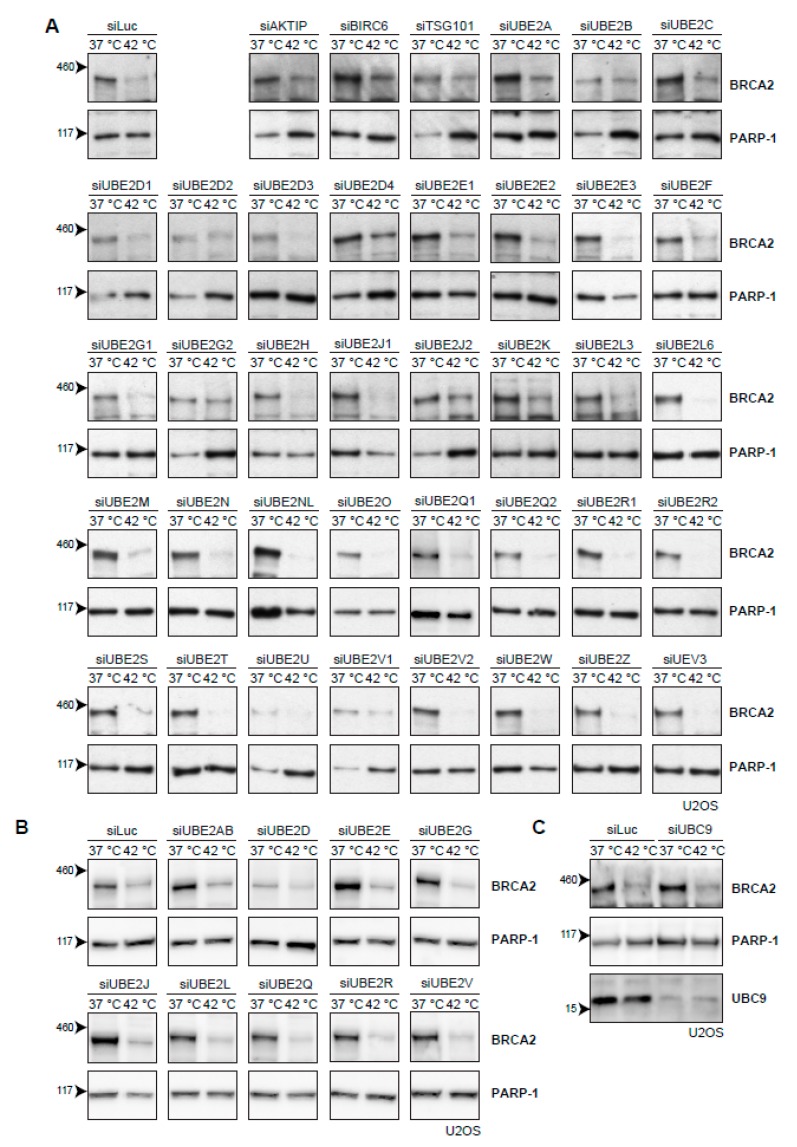
Efforts to identify the E2-conjugating enzyme that mediates degradation of BRCA2 upon hyperthermia. (**A**) Results of individual knock-down of all known ubiquitin E2-conjugating enzymes in U2OS cells. Efficiencies of E2 siRNAs were previously assessed by quantitative polymerase chain reaction PCR [37]. (**B**) Results of simultaneous knock-downs of E2-conjugating enzymes belonging to specific families in U2OS cells. (**C**) Results of knock-down of the small ubiquitin-like modifier (SUMO)-E2 enzyme Ubiquitin Conjugating Enzyme 9 (UBC9) in U2OS cells.

**Figure 5 cancers-11-00097-f005:**
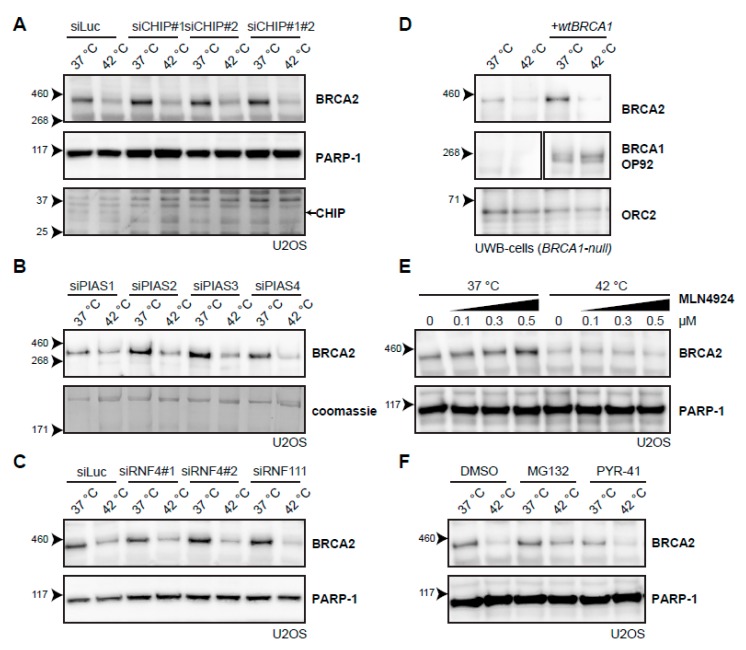
Searching for the E3-ligase that mediates degradation of BRCA2 upon hyperthermia. (**A**) Effect of hyperthermia on BRCA2 levels upon knock-down of the Heat-shock Protein (HSP)-associated E3-ligase, C terminus of HSC70-Interacting Protein (CHIP), in U2OS cells using two different siRNAs. Arrow on the right next to the CHIP panel indicates the predicted position of the CHIP protein. (**B**) Effect of hyperthermia on BRCA2 levels upon knock-down of the SUMO-E3 ligases Protein Inhibitor Of Activated STAT( PIAS) 1-4 in U2OS cells using siRNAs. (**C**) Effect of hyperthermia on BRCA2 levels upon siRNA knock-down of DNA-repair-related, SUMO-targeted ubiquitin ligases, Ring Finger (RNF) 4 and RNF111. (**D**) Detection of BRCA2 and BRCA1 in UWB1.289 (BRCA1-null) cells with or without complementation of wild type BRCA1 (wtBRCA1). (**E**) Detection of BRCA2 in U2OS cells treated with increased doses of the neddylation-inhibitor MLN4924, added 30 min prior to hyperthermia. (**F**) Detection of BRCA2 in U2OS cells treated with or without hyperthermia in the presence of proteasome inhibitor MG132 (10 µM) or the E1-activating enzyme inhibitor PYR-41 (10 µM). Inhibitors were added 1 h prior to hyperthermia treatment, and left on for the duration of the treatment.

**Figure 6 cancers-11-00097-f006:**
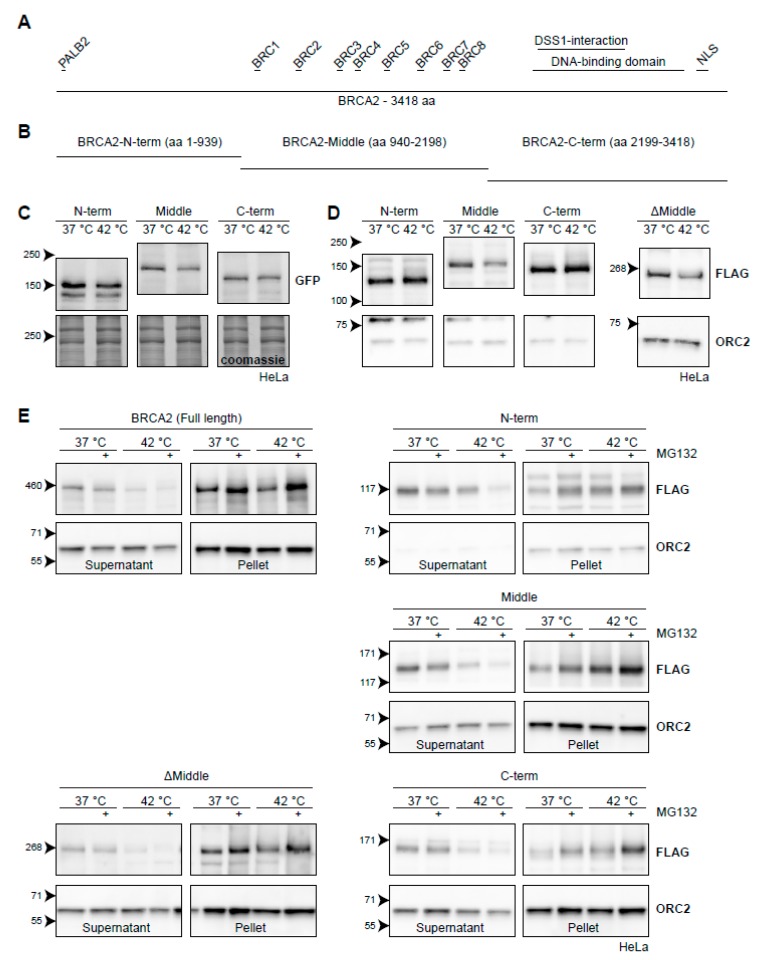
BRCA2 degradation upon heat treatment from the perspective of BRCA2. (**A**) Graphical representation of the full-length 3418-aa BRCA2-protein. Interaction domains are indicated as small bars above the full-length protein. (**B**) Arbitrary truncations of the BRCA2-protein to be tested for heat stability. (**C**) HeLa cells stably expressing GFP-BRCA2 fragments. (**D**) HeLa cells stably expressing FLAG-BRCA2 fragments. The right panel contains a FLAG-tagged ΔMiddle construct, which effectively is a fusion of the constructs designated as the N and C-termini. (**E**) Hela cells expressing the different constructs were subjected to hyperthermia in the absence or presence of 50 µM MG132 and processed following a fractionation protocol prior to lysis. The resulting supernatant and pellet fractions are shown.

**Figure 7 cancers-11-00097-f007:**
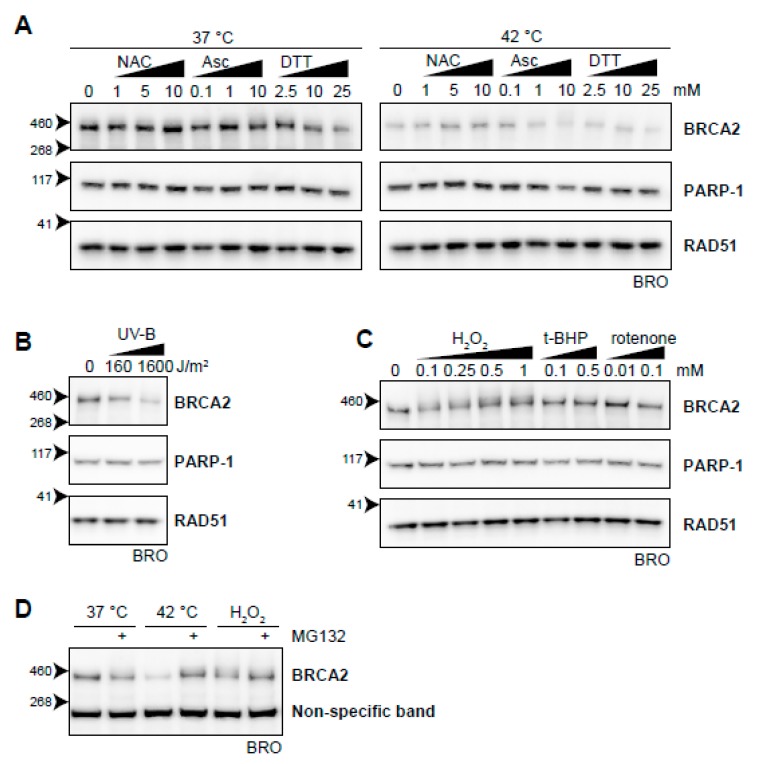
Oxidative stress as an inducer of BRCA2 degradation. **(A**) BRCA2 levels in BRO cells subjected to hyperthermia in the presence of indicated doses of antioxidant compounds N-acetylcysteine (NAC), ascorbic Acid (Asc), or dithiothreitol (DTT). (**B**) BRCA2 levels in BRO cells subjected to the indicated dose of ultraviolet-B and lysed three hours afterwards. (**C**) BRCA2 levels in BRO cells treated with various concentrations of oxidative stress-inducers H_2_O_2_, tert-Butyl hydroperoxide (t-BHP), and Rotenone. (**D**) BRCA2 levels in BRO cells exposed to hyperthermia or H_2_O_2_ (0.1 mM) in the presence or absence of proteasome inhibitor MG132 (50 µM).

**Table 1 cancers-11-00097-t001:** GFP nanobody bead pull-down from Brca2^GFP/GFP^ mES cells labelled for three-state Stable Isotope Labelling with Amino acids in Cell culture (SILAC). *Brca2^GFP/GFP^* mES cells were labelled with Light (L, K0R0), Medium (M, K4R6), and Heavy (H, K8R10) SILAC states. The cells were lysed in immunoprecipitation buffer immediately after the exposure to 42 °C for 0, 20, or 60 min. Precipitated proteins were separated by Sodium Dodecyl Sulfate-Polyacrylamide Gel Electrophoresis (SDS-PAGE) and analyzed by mass spectrometry. The treated to untreated SILAC ratios from two independent reciprocal label-swap experiments are given in the table, comparing 37 °C (“untreated”) with 20 min at 42 °C (“treated”) in the first column, and 37 °C with 60 min at 42 °C (“treated”) in the second.

Protein	Peptides	Coverage	Maxquant Pep	20 Min HT		60 Min HT	
				Exp 1	Exp 2	Exp 1	Exp 2
BRCA2	211	65	0.0E + 00	1.20	0.90	0.28	0.53
PALB2	36	44.5	0.0E + 00	1.49	0.92	0.30	0.37
RAD51	16	57.5	0.0E + 00	1.24	0.57	0.18	0.22
BRCA1	19	12.9	2.5E − 58	1.25	1.39	0.15	0.83
KEAP1	21	39.6	3.5E − 164	1.42	1.02	0.29	0.42
MORF4L1	17	61.9	1.1E − 271	1.44	0.89	0.29	0.42
MORF4L2	11	43.1	2.2E − 78	1.47	0.79	0.71	0.49
Ubiquitin	8	56.4	8.3E − 87	3.42	4.60	4.06	3.54
HSPB1	10	67.9	4.3E − 69	1.45	2.26	3.19	1.87
USP28	6	6	1.0E − 12	1.61	3.75	1.00	2.74
		%		*SILAC ratio treated to untreated*			

**Table 2 cancers-11-00097-t002:** List of constructs.

Name	BRCA2-region (Amino Acid)	Molecular Weight (kDa)	Vector	Pro-Motor	Code/Ref
FLAG-BRCA2	Met 1–Ile 3418	382	pGb-LPL	CAG	pAZ148
GFP-BRCA2	Met 1–Ile 3418	406	pGb-LPL	CAG	pAZ114 [26]
BRCA2-N-GFP	Met 1–Thr 939	131	pGb-LPL	CAG	pAZ108
BRCA2-M-GFP	Gln 940–Glu 2198	167	pGb-LPL	CAG	pAZ109
BRCA2-C-GFP	Thr 2199–Ile 3418	162	pGb-LPL	CAG	pAZ110
FLAG-BRCA2-N	Met 1–Thr 939	107	pGb-LPL	CMV	pAZ97
FLAG-BRCA2-M	Gln 940–Glu 2198	123	pGb-LPL	CMV	pAZ98
FLAG-BRCA2-C	Thr 2199–Ile 3418	118	pGb-LPL	CMV	pAZ104
FLAG-BRCA2-ΔM	Met 1–Thr 939 Thr 2199–Ile 3418	250	pGb-LPL	CAG	pAZ253
Clover-HSP90	n.a.	109	pGb-LPL	CAG	n.a.
FLAG-HSP90	n.a.	82	pGb-LPL	CMV	[99]

**Table 3 cancers-11-00097-t003:** List of oligonucleotides used in this study.

Set	Name	Sequence
1F	GA15N-F	5’-GCTCCTGGGCAACGTGCCTCGAGATGCCTATTGGATCCAAA GAGAGGCCAAC-3’
1R	GA15N-R	5’-TTGCTCACCATGGTGGCCTCGAGGGTTGCTTGTTTATCACCT GTGT-3’
2F	GA15M-F	5’-GCTCCTGGGCAACGTGCCTCGAGATGCAAGTGTCAATTAAAA AAGATTTGGTTTATGTTCTTGC-3’
2R	GA15M-R	5’-TTGCTCACCATGGTGGCCTCGAAAGTTTCAGTTTTACCAATTT CCATTTTTACGTT-3’
3F	GA15C-F	5’-GCTCCTGGGCAACGTGCCTCGAGATGACTTTTTCTGATGTTC CTGTGAAAACAAATATAGAAG-3’
3R	GA15C-R	5’-TTGCTCACCATGGTGGCCTCGAGGATATATTTTTTAGTTGTAA TTGTGTCCTGCTTATTTTTCTCACA-3’
4F	BRCA2-Nterm-F	5’-CAAGGATGACGACGACAAGAGCCCTATTGGATCCAAAGAGA GGC-3’
4R	BRCA2-Nterm-R	5’-GCTGATTATGATCTAGAGTCAGGTTGCTTGTTTATCACCTGT GTCT-3’
5F	GA04M-F	5’-CAAGGATGACGACGACAAGAGATCTACCCAAGTGTCAATTA AAAAAGATTTGGTTTATGT-3’
5R	GA04M-R	5’-GCTGATTATGATCTAGAGTCAGATCTTTTCAGTTTTACCAATT TCCATTTTTACGTTTTTAGGT-3’
6F	GA04C-F	5’-CAAGGATGACGACGACAAGAGATCTACTTTTTCTGATGTTCC TGTGAAAACAAATATAGAAG-3’
6R	GA04C-R	5’-GCTGATTATGATCTAGAGTCAGATCTTTAGATATATTTTTTAGT TGTAA TTGTGTCCTGCTTATTTTTCTCACAT-3’
7F	B2flFLAG-F1	5’-CCTGGGCAACGTGCCGATTATAAAGACCACGATGGAGACT ATAAAGATCATGACATTGACT-3’
7R	B2flFLAG-R2	5’-CACTGTCCTTCCTGCAGGCATGACAGAGAA-3’
8F	B2intdel-GA-F1	5’-TTCTCTGTCATGCCTGCAGGAAGGAC-3’
8R	B2intdel-GA-R1	5’-CAACATTTAAGTTATTTGATAATTTGGTTGCTTGTTTATCACC TGTGTCT-3’

**Table 4 cancers-11-00097-t004:** List of used chemicals.

Name	Reference/Suppliers
Bafilomycin	Sigma-Aldrich
NMS-873	Selleckchem
Ganetespib	Syntha Pharmaceuticals
MG132	Merck Millipore
Cycloheximide	Sigma-Aldrich
MLN4924	MedChem Express
PYR-41	Calbiochem
NAC	Sigma-Aldrich
Ascorbic Acid	Sigma-Aldrich
DTT	Sigma-Aldrich
H2O2	Sigma-Aldrich
t-BHP	Sigma-Aldrich
Rotenone	MP Biomedicals

**Table 5 cancers-11-00097-t005:** List of siRNAs.

siRNA	SenseSequence	Reference/Suppliers
Luc	CGUACGCGGAAUACUUCGA	Thermo Scientific
siAKTIP#1	GAAUUUACCUUGGUUGUGA	[37]
siAKTIP#2	AGAAAACAGUGGCGACUUA	[37]
siBIRC6#1	UCAUUGCCUUACUCACAUA	[37]
siBIRC6#2	GGUCAAAGAUCACUUAGUA	[37]
siTSG101#1	AGUAGCCGAGGUUGAUAAA	[37]
siTSG101#2	AAACUGAGAUGGCGGAUGA	[37]
siUBE2A#1	UGAUGUGUCUUCCAUUCUA	[37]
siUBE2A#2	GAUGAACCCAAUCCCAAUA	[37]
siUBE2B#1	AUAGACAACUGGUCUGUUA	[37]
siUBE2B#2	UUGGACCAGAAGGGACACC	[37]
siUBE2C#1	GCAAGAAACCUACUCAAAG	[37]
siUBE2C#2	UAAAUUAAGCCUCGGUUGA	[37]
siUBE2D1#1	UACUGUAUGUGUUGUCUAA	[37]
siUBE2D1#2	CAACAGACAUGCAAGAGAA	[37]
siUBE2D2#1	CAGUAAUGGCAGCAUUUGU	[37]
siUBE2D2#2	CCAACCAGAUUAAACUCUA	[37]
siUBE2D3#1	UGAUGUAAAGUUCGAAAGA	[37]
siUBE2D3#2	CCACAAUUAUGGGACCUAA	[37]
siUBE2D4#1	CAGCGUUGACUGUGUCAAA	[37]
siUBE2D4#2	GGAAUUAACCGACUUGCAG	[37]
siUBE2E1#1	GCGAUAACAUCUAUGAAUG	[37]
siUBE2E1#2	GGUGUAUUCUUUCUCGAUA	[37]
siUBE2E2#1	ACUUGAAAGAUUUGGGAUU	[37]
siUBE2E2#2	UCACCAGACUAUCCGUUUA	[37]
siUBE2E3#1	GCAUAGCCACUCAGUAUUU	[37]
siUBE2E3#2	GCUAAGUUAUCCACUAGUG	[37]
siUBE2F#1	GGAAUAAAGUGGAUGACUA	[37]
siUBE2F#2	CAACAUAAAUACAGCAAGA	[37]
siUBE2G1#1	UGUUGAUGCUGCGAAAGAA	[37]
siUBE2G1#2	GGGAAGAUAAGUAUGGUUA	[37]
siUBE2G2#1	AUGAUGACUUAAUGUCGAA	[37]
siUBE2G2#2	UGACGAAAGUGGAGCUAAC	[37]
siUBE2H#1	CGAGAGUAAACAUGAGGUU	[37]
siUBE2H#2	CUACUGAACUGUCGAAGGA	[37]
siUBE2J1#1	GAACUGGCUAGGCAAAUAA	[37]
siUBE2J1#2	GAAAGAAGCGGCAGAAUUG	[37]
siUBE2J2#1	GAAGGUGGCUAUUAUCAUG	[37]
siUBE2J2#2	GCACAAGACGAACUCAGUA	[37]
siUBE2K#1	CUCUCCGCACGGUAUUAUU	[37]
siUBE2K#2	GAAUCAAGCGGGAGUUCAA	[37]
siUBE2L3#1	UGAAGAGUUUACAAAGAAA	[37]
siUBE2L3#2	GGGCUGACCUAGCUGAAGA	[37]
siUBE2L6#1	UGAUCAAAUUCACAACCAA	[37]
siUBE2L6#2	UCAAUGUGCUGGUGAAUAG	[37]
siUBE2M#1	AGCCAGUCCUUACGAUAAA	[37]
siUBE2M#2	GAUGAGGGCUUCUACAAGA	[37]
siUBE2N#1	GCGGAGCAGUGGAAGACCA	[37]
siUBE2N#2	CUAUCUAGCUUGUGUGUCA	[37]
siUBE2NL#1	AAACGUGAACUAUUACUUG	[37]
siUBE2NL#2	GACAAGUUGGAAAGAAUAA	[37]
siUBE2O#1	ACAUCGACUGUGCCGUCAA	[37]
siUBE2O#2	GGGACUACAUUGCCUAUGA	[37]
siUBE2Q1#1	UCAUCUCCGACCUGUGUAA	[37]
siUBE2Q1#2	GAAAGGGAAUACUCUGCUA	[37]
siUBE2Q2#1	UACAGAUCACAGAGUUAUA	[37]
siUBE2Q2#2	GUAUGGAACUUCUCACAAA	[37]
siUBE2R1#1	CGCAGAACGUCAGGACCAU	[37]
siUBE2R1#2	GGAAGUGGAAAGAGAGCAA	[37]
siUBE2R2#1	UGUGAGGACUAUCCUAUUA	[37]
siUBE2R2#2	CCACAACCCUGGCGGAAUA	[37]
siUBE2S#1	AUGGCGAGAUCUGCGUCAA	[37]
siUBE2S#2	ACAAGGAGGUGACGACACU	[37]
siUBE2T#1	AGAGAGAGCUGCACAUGUU	[37]
siUBE2T#2	CCUGCGAGCUCAAAUAUUA	[37]
siUBE2U#1	ACAGGCCAUUACAAAUGAA	[37]
siUBE2U#2	GAAGUGGAAUACAAACUAU	[37]
siUBE2V1#1	GGACAGUGUUACAGCAAUU	[37]
siUBE2V1#2	GUGGAUGCAUACCGAAAUA	[37]
siUBE2V2#1	AGUUGUACUUCAAGAGCUA	[37]
siUBE2V2#2	GUUAAAGUUCCUCGUAAUU	[37]
siUBE2W#1	CGACCACCGGAUAAUUCUU	[37]
siUBE2W#2	GCGAACAUGUAACAAGAAU	[37]
siUBE2Z#1	AUGUUCGUUGUACCUGAUA	[37]
siUBE2Z#2	GGGAAAGUCUGCUUGAGUA	[37]
siUEV3#1	CGAUGGACCUUGAAAUCUU	[37]
siUEV3#2	AGAAAGACCUGCUGAAUUU	[37]
UBC9	GGGAUUGGUUUGGCAAGAA	[39]
CHIP#1	GCGCUCUUCGAAUCGCGAAGA	Invitrogen
CHIP#2	UGCCGCCACUAUCUGUGUAAU	Invitrogen
PIAS1	GGAUCAUUCUAGAGCUUUA	[43]
PIAS2	CUUGAAUAUUACAUCUUUA	[43]
PIAS3	CCCUGAUGUCACCAUGAAA	[43]
PIAS4	GGAGUAAGAGUGGACUGAA	[43]
RNF4#1	GAAUGGACGUCUCAUCGUU	[39]
RNF4#2	GACAGAGACGUAUAUGUGA	Thermo Scientific
RNF111	GGAUAUUAAUGCAGAGGAA	[39]

**Table 6 cancers-11-00097-t006:** List of antibodies.

Host	Epitope	Dilutions Used	Reference/Suppliers
Mouse	BRCA2 (OP95)	WB 1:1000	EMD Millipore
Mouse	FLAG (M2)	WB 1:1000–1:5000	Sigma-Aldrich
Mouse	ORC2 (68348)	WB 1:1000	Abcam
Rabbit	Brca2 (27976)	WB 1:500	Abcam
Mouse	GFP (clones 7.1 and 13.1)	WB 1:1000–1:5000	Sigma-Aldrich
Mouse	PARP-1 (C2-10)	WB 1:5000	Enzo Lifesciences
Mouse	BRCA1 (OP92)	WB 1:250	EMD Millipore
Rabbit	Cyclin A (C-19)	WB 1:5000	Santa Cruz
Goat	RAD54 (D-18)	WB 1:1000	Santa Cruz
Rabbit	RAD51 (2307)	WB 1:10,000	Home-made [100]
Rabbit	RAD51 (2308)	IF 1:10,000	Home-made [100]
Mouse	GRB2	WB 1:1000	BD Pharmingen
Mouse	HSP90 (AC88, 13492)	WB 1:5000	Abcam
Rabbit	CDC37 (3618S)	WB 1:1000	Cell Signaling
Goat	UBC9 (N-15)	WB 1:1000	Santa Cruz
Rabbit	CHIP (PA1-015)	WB 1:1000	Thermo Scientific
Sheep	Peroxidase anti-Mouse IgG (H+L)	WB 1:2000	Jackson Immunoresearch
Donkey	Peroxidase anti-Rabbit IgG (H+L)	WB 1:2000	Jackson Immunoresearch
Donkey	Peroxidase anti-Goat IgG (H+L)	WB 1:2000	Jackson Immunoresearch
Goat	Alexa-Fluor-594 (red)	IF 1:1000	Thermo Scientific

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
