# Peer review of "On the Mechanism of Hyperthermia-Induced BRCA2 Protein Degradation"

_cancers, 2019, doi:10.3390/cancers11010097_

Reviewer 1 Report

In the present paper the authors continue their research on the mechanisms of hyperthermia induced radiosensitization. Hyperthermia is a tumor treatment modality that received in the past considerable attention either for use alone, or in combination with ionizing radiation (IR). Particularly the combination with IR generated excitement because hyperthermia was found to be a strong radiosensitizer, specifically for cells in radioresistant phases of the cell cycle. Despite strong evidence at the cellular level, however, the molecular mechanism(s) of radiosensitization, particularly of mild hyperthermia, typically defined as temperatures below 42.5 °C, had remained elusive. A widely held model was that radiosensitization is multiparametric and results from the unfolding of proteins involved in the repair of IR induced DNA double strand breaks (DSB).

In a breakthrough paper a few years ago the laboratory of Dr. Kanaar demonstrated that mild hyperthermia radiosensitizes cells by mainly inhibiting homologous recombination repair (HRR). Notably, this effect appeared to be induced by the rather specific degradation of the tumor suppressor protein BRCA2 that is essential for the formation of the Rad51-DNA filament during HRR. Here, Dr. Kanaar takes this work further and reports that hyperthermia-induced BRCA2 degradation is evolutionarily conserved and depends on HSP90. Results are also presented suggesting that ubiquitin may not be involved in directly targeting BRCA2 for degradation via the proteasome, and that BRCA2 degradation is modulated by oxidative stress and radical scavengers.

The paper is well written and contains a fair amount of new data that advance our understanding of the process. Although the wealth of results presented do not unequivocally define the mechanism of degradation, they inform the directions further research should focus on and will be of interest to the readers of the Journal. Elucidation of the mechanisms underpinning heat induced radiosensitization should help to improve the clinical application of this treatment modality in the therapy of human cancer.

Specific Comments

1.       Figures 1A and B show that HeLa BRAC2 as well as murine BRCA2 and murine BRCA2-GFP are degraded upon hyperthermia, suggesting that the previously reported response is reproducible in additional cell lines and evolutionarily conserved. These results are straightforward.

2.       Figure 1C shows similar results with BRCA1 using three different antibodies. The results shown do not allow the generation of a consistent picture regarding the response to hyperthermia of this protein and need further discussion and clarifications (band-identities/intensities). Alternatively, the authors could consider removing these data from the paper as they appear secondary to its message.

3.       Figure 2A shows that the proteasome inhibitor MG132 reduces hyperthermia induced BRCA2 degradation. Can the authors comment as to why this protection diminishes with increasing drug dose?

4.       Figure 2D: It is not clear from the results shown that inhibition of HSP90 enhances hyperthermia induced BRCA2 degradation, particularly after considering the indicated loading control.

5.       Figure 4A: Occasionally, as for example for UBE2B the results shown are equivocal! Can the authors comment?

6.       Figures 4 and 5 summarize results showing lack of protection of BRCA2 degradation after exposure to hyperthermia by inhibiting E2 conjugating or E3 ligases. Even inhibition of the E1-activating enzyme fails to protect, although MG132 does protect. This sort of conundrum may come from non-specific effects of MG132. The authors should use more specific next generation proteasome inhibitors such as bortezomib to attempt resolving this issue.

7.       Figure 8A: The effects of NAC or Ascorbic acid are small and this should be acknowledged.

Author Response

Thank you for your quick reply on our submitted manuscript “On the mechanism of hyperthermia-induced BRCA2 protein degradation”. In this rebuttal letter we will address the reviewer's questions and suggestions point by point. 

Reviewer 1

In the present paper the authors continue their research on the mechanisms of hyperthermia induced radiosensitization. Hyperthermia is a tumor treatment modality that received in the past considerable attention either for use alone, or in combination with ionizing radiation (IR). Particularly the combination with IR generated excitement because hyperthermia was found to be a strong radiosensitizer, specifically for cells in radioresistant phases of the cell cycle. Despite strong evidence at the cellular level, however, the molecular mechanism(s) of radiosensitization, particularly of mild hyperthermia, typically defined as temperatures below 42.5 °C, had remained elusive. A widely held model was that radiosensitization is multiparametric and results from the unfolding of proteins involved in the repair of IR induced DNA double strand breaks (DSB).

In a breakthrough paper a few years ago the laboratory of Dr. Kanaar demonstrated that mild hyperthermia radiosensitizes cells by mainly inhibiting homologous recombination repair (HRR). Notably, this effect appeared to be induced by the rather specific degradation of the tumor suppressor protein BRCA2 that is essential for the formation of the Rad51-DNA filament during HRR. Here, Dr. Kanaar takes this work further and reports that hyperthermia-induced BRCA2 degradation is evolutionarily conserved and depends on HSP90. Results are also presented suggesting that ubiquitin may not be involved in directly targeting BRCA2 for degradation via the proteasome, and that BRCA2 degradation is modulated by oxidative stress and radical scavengers.

The paper is well written and contains a fair amount of new data that advance our understanding of the process. Although the wealth of results presented do not unequivocally define the mechanism of degradation, they inform the directions further research should focus on and will be of interest to the readers of the Journal. Elucidation of the mechanisms underpinning heat induced radiosensitization should help to improve the clinical application of this treatment modality in the therapy of human cancer.

We would like to thank Reviewer 1 for the kind words.

Specific Comments

1.  Figures 1A and B show that HeLa BRAC2 as well as murine BRCA2 and murine BRCA2-GFP are degraded upon hyperthermia, suggesting that the previously reported response is reproducible in additional cell lines and evolutionarily conserved. These results are straightforward.

Indeed, they are. Since we use mouse BRCA2 fused to GFP in the mass spec experiment shown in Figure 7, we did need to confirm that mouse BRCA2-GFP is indeed also degraded by hyperthermia.

2.  Figure 1C shows similar results with BRCA1 using three different antibodies. The results shown do not allow the generation of a consistent picture regarding the response to hyperthermia of this protein and need further discussion and clarifications (band-identities/intensities). Alternatively, the authors could consider removing these data from the paper as they appear secondary to its message.

We agree with the reviewer that the different antibodies, generated against different epitopes of BRCA1, do give results whose interpretation is currently not straightforward. Furthermore, the BRCA1 experiment is only tangential to this manuscript. Therefore, we follow the reviewer’s suggestion to remove it from the manuscript. 

3.  Figure 2A shows that the proteasome inhibitor MG132 reduces hyperthermia induced BRCA2 degradation. Can the authors comment as to why this protection diminishes with increasing drug dose?

We have done this experiment many times with the highest dose, and always found that it fully protected heat-induced degradation of BRCA2. In the blot shown, this is not particularly obvious, since in this particular instance there might be a difference in loading. Given the requested response time of five days we could not redo the blot, but are willing to do so if allowed the time.

4.  Figure 2D: It is not clear from the results shown that inhibition of HSP90 enhances hyperthermia induced BRCA2 degradation, particularly after considering the indicated loading control.

The choice of loading control was a bit unfortunate in this case, as it might be that cyclin A, a highly dynamic protein in sense of degradation and synthetisation, is also a client of HSP90. While qualitatively the same as our previously published work, this makes the result quantitatively less impressive. However, it is certainly real, as can be seen for example in Vriend and van den Tempel et al, Oncotarget 2017, doi: 10.18632/oncotarget.22142, Figure 1C.

5.  Figure 4A: Occasionally, as for example for UBE2B the results shown are equivocal! Can the authors comment?

Due to the limited amount of time spent in the laboratory of Prof Steve Jackson in Cambridge, where the lead author performed these experiments, we were unable to reload samples to compensate for differences in the loading controls. To solve this problem, we have quantified all samples relative to the loading control and found no degradation in any of them. To keep the figure as small as possible, we have omitted these quantification results, but have now mentioned in the text we have performed this step. 

6.  Figures 4 and 5 summarize results showing lack of protection of BRCA2 degradation after exposure to hyperthermia by inhibiting E2 conjugating or E3 ligases. Even inhibition of the E1-activating enzyme fails to protect, although MG132 does protect. This sort of conundrum may come from non-specific effects of MG132. The authors should use more specific next generation proteasome inhibitors such as bortezomib to attempt resolving this issue.

We have added an additional line acknowledging the non-specific effects of MG132 on page 15: “Alternatively, protection of hyperthermia-mediated degradation of BRCA2 by MG132 could be due to off targets effects of the inhibitor. However, epoxomicin, an alternative proteasome inhibitor, yielded similar results to MG132.”

7.  Figure 8A: The effects of NAC or Ascorbic acid are small and this should be acknowledged.

The reviewer is right and we have acknowledged this now at the specific section. 

Reviewer 2 Report

The manuscript by Van den Tempel et al. investigates the underlying mechanisms of hyperthermia induced degradation of the BRCA2 protein. This is very relevant topic since recently it was shown that hyperthermia interferes with the DNA damage response and thereby creates new opportunities for cancer treatments. In general, the experiments described in the manuscript are well performed, though I have a couple of questions regarding the interpretation of the results.

-          Could the authors explain the concentration dependent effect of MG132 on BRCA2 degradation at 37 degrees shown in figure 2A? A significant decrease of BRCA2 can be observed for 25 uM MG132 at 37 degrees.

-          In line 127-128 the authors state that “inhibition of HSP90 indeed enhanced BRCA2-degradation upon hyperthermia”. However, in my opinion no significant differences between 0 mM ganetespib and the other concentrations of ganetespib at 42 degrees can be observed in figure 2D. In all cases the BRCA2 is completely degraded. Therefore the role of HSP90 in hyperthermia-induced BRCA2 degradation remains unclear and needs additional experiments.

-          Why are the experiments with various inhibitors (figure 2 & 3) performed with different cell lines? Does one inhibitor give the same result for different cell lines?

-          According to the authors the experiment combining the HSP inhibitor (ganetespib), translation inhibitor (cycloheximide) and hyperthermia (i.e. results presented in figure 3) show that “HSP90 protects BRCA2 under hyperthermic conditions”.  Though what I miss to draw this conclusion is prove that cycloheximide indeed inhibits HSP90 activity (e.g. by looking at HSP72 production?). Their results only show that cycloheximide does not influence HSP90 production, although this might be expected.

-          The conclusions drawn based on the results presented in figure 7 are not very strong due to limited quality of the results: i) experiments are only performed in doublet, ii) there is a large variation between exp 1 and 2 and iii) the authors only show ratios and not the absolute values, which are also very relevant.

-          In line 329 the authors mention that the addition of H2O2 also led to BRCA2 degradation. H2O2 is also formed in water/tissue by ionizing radiation. This would mean that ionizing radiation directly interferes with DNA damage response. Could the authors comment on this statement? Is it correct? And what implications would this have?

And some minor remarks

-          Below figure 2B ‘U2OS’ is indicated as cell line, whereas in the caption HeLa cells are described for this panel

-          At line 273 it is unclear if this the continuation of the caption of figure 6 or a new paragraph in the text

Author Response

Thank you for your quick reply on our submitted manuscript “On the mechanism of hyperthermia-induced BRCA2 protein degradation”. In this rebuttal letter we will address the reviewer’s questions and suggestions point by point. 

Reviewer 2

Comments and Suggestions for Authors

The manuscript by Van den Tempel et al. investigates the underlying mechanisms of hyperthermia induced degradation of the BRCA2 protein. This is very relevant topic since recently it was shown that hyperthermia interferes with the DNA damage response and thereby creates new opportunities for cancer treatments. In general, the experiments described in the manuscript are well performed, though I have a couple of questions regarding the interpretation of the results.

We would like to thank reviewer 2 for acknowledging the merit of this particular manuscript and the kind words.  

Could the authors explain the concentration dependent effect of MG132 on BRCA2 degradation at 37 degrees shown in figure 2A? A significant decrease of BRCA2 can be observed for 25 uM MG132 at 37 degrees.

In line 127-128 the authors state that “inhibition of HSP90 indeed enhanced BRCA2-degradation upon hyperthermia”. However, in my opinion no significant differences between 0 mM ganetespib and the other concentrations of ganetespib at 42 degrees can be observed in figure 2D. In all cases the BRCA2 is completely degraded. Therefore the role of HSP90 in hyperthermia-induced BRCA2 degradation remains unclear and needs additional experiments.

These two points were also made by reviewer 1 (point 3 and 4), and are addressed there. 

Why are the experiments with various inhibitors (figure 2 & 3) performed with different cell lines? Does one inhibitor give the same result for different cell lines?

This work is a collection of results gathered for >5 years. We have tested many of the inhibitors in various cell lines over the years. We have chosen to present the most elaborate results in this manuscript, taking along various concentrations. We have verified these results in at least one other cell line and have thus far not found inhibitors which behaves differently in different cell lines. 

According to the authors the experiment combining the HSP inhibitor (ganetespib), translation inhibitor (cycloheximide) and hyperthermia (i.e. results presented in figure 3) show that “HSP90 protects BRCA2 under hyperthermic conditions”.  Though what I miss to draw this conclusion is prove that cycloheximide indeed inhibits HSP90 activity (e.g. by looking at HSP72 production?). Their results only show that cycloheximide does not influence HSP90 production, although this might be expected.

This is an interesting point, which we have not considered thus far. However, we feel our conclusions are still valid if we follow our previous reasoning. Cycloheximide is present for 2 hours during the experiment- one hour prior to hyperthermia, one hour during hyperthermia. In these 2 hours, translation is shut down. However, many proteins have a half-life extending 2 hours (including HSP90), and therefore, we do not expect an effect on HSP90 function in the experimental time frame.

The conclusions drawn based on the results presented in figure 7 are not very strong due to limited quality of the results: i) experiments are only performed in doublet, ii) there is a large variation between exp 1 and 2 and iii) the authors only show ratios and not the absolute values, which are also very relevant.

We agree with the reviewer that the data presented in figure 7 can be further extended. However, we want to note that the reciprocal label-swap experiments we performed are based on robust biological replicates, as cells that undergo the same treatment grow separately for ~2 weeks during the labelling stage, and the pull-downs are performed separately. This experimental setup is considered the gold standard for semi-quantitative SILAC-based proteomics. We also note that the variability in the measured SILAC ratios is comparable to that observed by immunoblotting. We note that the table reports the posterior error probabilities (PEP), that supported its reliability of protein identification, as well as the peptide numbers. 

In line 329 the authors mention that the addition of H2O2 also led to BRCA2 degradation. H2O2 is also formed in water/tissue by ionizing radiation. This would mean that ionizing radiation directly interferes with DNA damage response. Could the authors comment on this statement? Is it correct? And what implications would this have?

This is a very interesting point. Radiotherapy indeed may form H2O2, via initial formation of OH-. However, we expect the amounts we add to the cells here only to occur when irradiated with extremely high doses, which will most likely not occur in the clinic, and are therefore unlikely to be relevant. Of note, we have not found BRCA2 degradation upon gamma-irradiation between 2-4 Gy before. However, it will be interesting to see how the formation of peroxide by radiation relates to the formation by hyperthermia in a follow up study. 

And some minor remarks

Below figure 2B ‘U2OS’ is indicated as cell line, whereas in the caption HeLa cells are described for this panel

We thank the reviewer for pointing this out, and we have changed the figure caption. 

At line 273 it is unclear if this the continuation of the caption of figure 6 or a new paragraph in the text

We agree that this is an odd place to interrupt the flow of the paragraph and have now continued it until the following figure.